# β-adrenergic receptor signaling evokes the PKA-ASK axis in mature brown adipocytes

**Kazuki Hattori**[1¤]*, **Hiroaki Wakatsuki**[1], **Chihiro Sakauchi**[1], **Shotaro Furutani**[1], **Sho Sugawara**[1], **Tomohisa Hatta**[2], **Tohru Natsume**[2], **Hidenori Ichijo**[1]*

**1** The Laboratory of Cell Signaling, Graduate School of Pharmaceutical Sciences, The University of Tokyo, Tokyo, Japan, **2** Molecular Profiling Research Center for Drug Discovery (molprof), National Institute of Advanced Industrial Science and Technology (AIST), Tokyo, Japan

¤ Current address: Research Center for Advanced Science and Technology, The University of Tokyo, Komaba, Meguro-ku, Tokyo, Japan
* kzkhattori@13.alumni.u-tokyo.ac.jp (KH); ichijo@mol.f.u-tokyo.ac.jp (HI)

**Data Availability Statement:** All relevant data are within the manuscript and its Supporting Information files.

**Funding:** This work was supported by a Grant-in-Aid for Scientific Research (KAKENHI) from the

## Abstract

Boosting energy expenditure by harnessing the activity of brown adipocytes is a promising strategy for combatting the global epidemic of obesity. Many studies have revealed that the β$_3$-adrenergic receptor agonist is a potent activator of brown adipocytes, even in humans, and PKA and p38 MAPK have been demonstrated for regulating the transcription of a wide range of critical genes such as *Ucp1*. We previously revealed that the PKA-ASK1-p38 axis is activated in immature brown adipocytes and contributes to functional maturation. However, the downstream mechanisms of PKA that initiate the p38 MAPK cascade are still mostly unknown in mature brown adipocytes. Here, we identified the ASK family as a crucial signaling molecule bridging PKA and MAPK in mature brown adipocytes. Mechanistically, the phosphorylation of ASK1 at threonine 99 and serine 993 is critical in PKA-dependent ASK1 activation. Additionally, PKA also activates ASK2, which contributes to MAPK regulation. These lines of evidence provide new details for tailoring a βAR-dependent brown adipocyte activation strategy.

## Introduction

The global obesity epidemic shows no sign of stopping [1, 2]. Since obesity is closely linked to multiple serious complications, such as heart diseases and diabetes [3], humankind should undoubtedly tackle this issue to extend healthy life expectancy. Although at least some cases of obesity are preventable [4], identifying methods for reversing the bodyweight of existing patients is essential in terms of patient care. Simply stated, we have two strategies for reducing body weight: suppressing energy intake and enhancing energy expenditure. Currently approved drugs mainly employ the former approach, i.e., reducing appetite or suppressing energy uptake in the intestine [3]. Notably, these drugs all have potentially severe side effects, which limits their versatility. Many studies have focused on the latter strategy, i.e., boosting energy expenditure, but it has not yet been successful. Since mitochondria are the organelles that oxidize high amounts of substrates to produce energy, it is logical to focus on tissues with high mitochondrial activity or that are rich in mitochondria to boost energy expenditure.

Japan Society for the Promotion of Science (JSPS; grant numbers JP18H03995 and JP25221302 to H.I., JP16K18872 and JP25893036 to K.H., see https://kaken.nii.ac.jp/ for details), Project for Elucidating and Controlling Mechanisms of Aging and Longevity from Japan Agency for Medical Research and Development (AMED; grant number JP19gm5010001 to H.I.), and the Kowa Life Science Foundation (2016 A-2 to K.H.). The funders had no role in study design, data collection and analysis, decision to publish, or preparation of the manuscript.

**Competing interests:** The authors have declared that no competing interests exist.

However, electron transport chain activity is usually coupled with ATP production, and the abundance of ATP is tightly regulated, which may limit the maximum energy consumption [5]. Brown adipose tissue (BAT) is a unique tissue rich in mitochondria that selectively expresses uncoupling protein 1 (UCP1), a mitochondria inner membrane-resident uncoupler. Activated UCP1 induces energy-consuming futile cycles and dissipates energy in the heat without producing ATP [6]. Many proof-of-concept experiments have revealed that causing uncoupling is beneficial for counteracting obesity [7]. Therefore, methods for increasing the level and/or activity of UCP1$^+$ adipocytes (for another type of UCP1$^+$ cell, beige adipocytes, see this review [8]) have attracted attention in the past decade. One promising strategy is the administration of β$_3$-adrenergic receptor (β$_3$AR) agonists, which both induce the temporal activation of UCP1 and the expression of UCP1 in rodent models [9–11]. A human study utilizing an approved β$_3$AR agonist, mirabegron, reported mildly increased BAT activity [12–14] and improved glucose clearance [13, 15], suggesting that the β$_3$AR axis is a promising target for developing novel therapeutic strategies. However, many studies used a 2–4 times higher dose than the approved dose, and the approved dose exhibits only limited effects; hence, off-target effects of β$_3$AR agonists and cross-activation of β$_1$AR and β$_2$AR should be considered. Clinically, constructing selective methods for activating β$_3$AR on brown and/or beige adipocytes is crucial, but it is of critical importance to understand the signaling mechanisms induced by β$_3$AR agonists in mature brown adipocytes to fully understand the potential side effects of β$_3$AR agonism. However, a limited number of studies have uncovered the involvement of several kinases in β$_3$AR signaling [16–19]; specifically, the downstream targets of cAMP-dependent protein kinase (PKA) are not fully known.

Among a wide variety of signaling pathways, the stress-responsive mitogen-activated protein kinase (MAPK) pathway regulates multiple functions such as gene expression and cell death depending on cell types [20, 21]. Canonically, three-tiered cascades, MAPK kinase kinase (MAP3K)-MAPK kinase (MAP2K)-MAPK, are the central components of the MAPK pathway, and a tremendous variety of signaling is converged on two exclusive MAPKs, namely, p38s and c-Jun N-terminal kinases (JNKs) [21]. The apoptosis signal-regulating kinase (ASK) family, which consists of ASK1, ASK2, and ASK3, is a member of MAP3K and shares similar amino acid sequences throughout the kinase domain but functions differently in many cases [22–24]. Additionally, ASKs can form heteromeric complexes and are mutually interactive in some cases [25–28]. Although the key role of ASK1 is cell death regulation [29–33], we previously reported that ASK1 also functions as a differentiation regulator through gene induction in immature brown adipocytes and contributes to functional maturation [9]. However, there have been no reports regarding the functions of the ASK family in mature brown adipocytes. Importantly, cell signaling patterns are incredibly diverse and depend on many parameters, such as cell type, stimulus type, stimulus intensities, and species. Hence, we can assume that immature and mature adipocytes share similar signaling pathways but cannot assure this unless we actually examine it. Furthermore, the PKA-dependent ASK1 activation mechanism and the other ASK family involvement in adipocyte signaling were unknown.

Here, we show that ASK1 and ASK2 are essential kinases that function as the hub of PKA and MAPK signaling in mature brown adipocytes through the newly identified phosphorylation on ASK1.

# Materials and methods

## Antibodies and reagents

Anti-phospho-ASK (mouse ASK1: Thr 845, mouse ASK2: Thr 807) [34], anti-mouse ASK2 [25], anti-human ASK2 [25] and anti-human ASK3 [35] polyclonal antibodies were generated

as previously described. Phospho-specific antibodies against HSL (Ser 660) (#4126), p38 (Thr 180/Tyr 182) (#9211 or #4511), JNK (Thr 183/Tyr 185) (#9251), and ASK1 (Ser 967) (#3764) were purchased from Cell Signaling Technology. An anti-p38α antibody (#9228), an anti-HSL antibody (#4107) and an anti-Lipin1 antibody (#5195) were also purchased from Cell Signaling Technology. An anti-ASK1 antibody (ab45178), an anti-VDAC antibody (ab14734) and an anti-UCP1 antibody (ab10983) were purchased from Abcam, an anti-GFP antibody (M048-3) was purchased from MBL, an anti-HA-tag antibody (#11867431001) was purchased from Roche, an anti-Flag-tag antibody (#012–22384) was purchased from Wako, an anti-actin antibody (A3853) was purchased from Sigma Aldrich, an anti-α-tubulin antibody (MCA77G) was purchased from Bio-Rad, an anti-UQCRC1 antibody (#459140) was purchased from Thermo Fisher, an anti- Cytochrome C (CYCS) antibody (#556433) was purchased from BD Biosciences, and an anti-JNK1 antibody (sc-571) was purchased from Santa Cruz Biotechnology. Secondary antibodies (HRP-linked anti-rabbit IgG (#7074), HRP-linked anti-rat IgG (#7077), and HRP-linked anti-mouse IgG (#7076)) were purchased from Cell Signaling Technology. CL316,243 (sc-203895) was purchased from Santa Cruz Biotechnology. Forskolin was purchased from Sigma-Aldrich (F6886) or Tokyo Chemical Industry (F0855). Isoproterenol (I6504) and norepinephrine (A9512) were purchased from Sigma-Aldrich. H89 was purchased from Cell Signaling Technology (#9844S) or Cayman Chemical (#10010556). 8-pCPT-cAMP (#039–18121) and $H_2O_2$ (#081–04215) were purchased from Wako. An ASK1 inhibitor, K811, was kindly provided by Kyowa Hakko Kirin.

## Cell culture

HEK293A cells were purchased from Invitrogen and cultured in high-glucose DMEM (D5796, Sigma-Aldrich) containing 10% FBS in a 5% $CO_2$ atmosphere at 37˚C. Previously described methods were used for the isolation and culture of brown adipocytes [9]. Cell images were acquired by an inverted contrasting microscope (DM IL HC, Leica) with a digital microscope camera (RA-MC120HD, Leica). The medium of the brown adipocytes was replaced with basal culture medium (high-glucose DMEM containing 20% FBS) 1–3 h before stimulation. ASK1, ASK2, and ASK3 triple knockout HEK293A cells were established by the CRISPR-Cas9 system (see below for details) and cultured in the same way as the parent HEK293A cells.

## Transfection of plasmids

Plasmid transfections were conducted by mixing DMEM and PEI-Max (Polysciences) at room temperature and incubating for 5 min. The plasmid solution was then mixed with the transfection reagent solution and incubated for an additional 10–15 min before being added to the cells. The cells were incubated for two days before lysis.

Mouse PKACA (NM_008854.5) and PKACB (NM_011100.4) from cDNA prepared from the interscapular brown adipose tissue of wild-type mice were cloned into a pcDNA3 or pcDNA3/GW vector; these plasmids were established in our previous study [9]. Kinase-negative mutants of PKA (K72R and K72A) were generated by site-directed mutagenesis; these plasmids were also established in the previous study [9]. PKACB fragments (NT, CT, and KD) were generated by standard PCR methods using wild-type PKACB as a template and cloned into the pcDNA3/GW vector. Single amino acid substitution of mouse ASK1 was achieved by site-directed mutagenesis; other mouse ASK1 and ASK2 mutants were established in previous studies [25, 34, 36]. The amino acid sequence of mouse ASK1 was different from that of NP_032606.4 at 203 aa (K to R).

## Adenoviral infection

Mouse PKIα, WT mouse ASK1, mouse ASK1 S973A, and mouse ASK2 were cloned into the pAd/CMV/V5 vector with tags, and the virus was produced in HEK293A cells as described by

the manufacturer (Invitrogen). Immature brown adipocytes were treated with the viruses on day 2.

## Western blotting

Cells were lysed with IP lysis buffer (20 mM Tris-HCl, pH 7.5, 150 mM NaCl, 4 mM EDTA, pH 8.0, 1% w/v sodium deoxycholate, 1% v/v Triton X-100, 1 mM phenylmethylsulfonyl fluoride, 5 μg/mL leupeptin, 8 mM NaF, 1 mM $Na_3VO_4$, 12 mM β-glycerophosphate, 1.2 mM $Na_2MoO_4$, 5 μM cantharidin, and 2 mM imidazole). For the immunoprecipitation (IP) assay, anti-Flag beads (#016–22784, Wako) were added after the lysate sample was collected, and the mixture was incubated for 10 min at 4˚C. The gel was washed with IP lysis buffer three times. Both the lysate and IP samples were mixed with 2x sampling buffer (80 mM Tris-HCl, pH 8.8, 0.008% w/v bromophenol blue, 28.8% glycerol, 4% SDS, and 10 mM dithiothreitol) and boiled at 98˚C for 3 min. The prepared samples were resolved by SDS-PAGE and electroblotted onto the BioTrace PVDF membrane (Pall), the FluoroTrans W PVDF membrane (Pall), or the Immobilon-P PVDF membrane (IPVH00010, Millipore). The membranes were blocked with 2% skim milk (Yukijirushi) in TBS-T (50 mM Tris-HCl, 150 mM NaCl and 0.05% Tween 20, pH 8.0) and then probed with appropriate primary antibodies in TBS-T containing 5% bovine serum albumin (A001, Iwai Chemicals) and 0.1% w/v sodium azide (#31233, Nacalai Tesque). The membranes were washed and then probed with appropriate secondary antibodies in 2% skim milk. Antibody-antigen complexes were detected on X-ray film (Fujifilm) or by FUSION SOLO.7S.EDGE (Vilber), using an ECL system. The obtained films were scanned by a scanner and processed by ImageJ [37]. "ASK1 activation magnitude by PKA coexpression" in Fig 3 was defined as ($[phospho\text{-}ASK]_{with\ PKA}$ / $[total\text{-}ASK1]_{with\ PKA}$) / ($[phospho\text{-}ASK]_{without\ PKA}$ / $[total\text{-}ASK1]_{without\ PKA}$), and "basal activity of ASK1 mutants relative to WT" in Fig 3 was defined as ($[phospho\text{-}ASK]_{mutant,\ without\ PKA}$ / $[total\text{-}ASK1]_{mutant,\ without\ PKA}$) / ($[phospho\text{-}ASK]_{WT,\ without\ PKA}$ / $[total\text{-}ASK1]_{WT,\ without\ PKA}$). Representative data are shown for all Western blotting analyses, and more than two additional experimental replicates showed similar results.

## qRT-PCR

RNA was extracted using FastGene RNA Basic Kit (FG-80050, NIPPON Genetics) and reverse-transcribed with ReverTra Ace qPCR RT Master Mix with gDNA Remover (Toyobo, FSQ-301), following the manufacturer's instructions. Each cDNA was analyzed by QuantStudio 1 Real-Time PCR System (A40425, Thermo Fisher) using KAPA SYBR FAST qPCR Kits (#07959397001, Roche). Data were normalized to Hprt1. Primers were designed using the Universal Probe Library Assay Design Center (Roche). See the S1 Table for the primer sequences.

## Glucose uptake assay

Glucose uptake levels were assessed by Glucose Uptake-Glo™ Assay (J1341, Promega), and each value was normalized to the ATP levels, measured by Cell ATP Assay reagent Ver. 2 (#381–09301, Wako), according to the manufacturer's protocol. 10 nM insulin was treated for 30 min before measuring glucose uptake.

## Establishment of ASK1, ASK2, and ASK3 triple knockout cells

sgRNAs targeting human ASK1, ASK2, and ASK3 were cloned into pSpCas9(BB)-2A-Puro (PX459) V2.0 and transfected into HEK293A cells simultaneously [38]. The transfected cells

were selected with 0.5 μg/mL puromycin (A11138-03, Gibco), and the surviving cells were isolated to obtain monoclonal cells by limiting dilution. Cell lysates were prepared, and protein expression levels were assessed by Western blotting analyses. A cell line with no detectable ASK1, ASK2, or ASK3 expression was selected as a triple knockout cell line. Control cells were transfected with an intact vector, selected, and cloned in the same way. pSpCas9(BB)-2A-Puro (PX459) V2.0 (Addgene plasmid #62988; http://n2t.net/addgene:62988; RRID: Addgene_62988) was a gift from Feng Zhang. The DNA sequences that were cloned into the PX459 plasmid are as follows:

human ASK1 exon 1: 5'-CACCGCGGGGGCAGCCGACGGACCA-3'
human ASK2 exon 10: 5'-CACCGCCTGCAAAGCTCGAGGTTCG-3'
human ASK3 exon 2: 5'-CACCGAGCTTTCTCGGACTCCAAGA-3'

## Oil red O staining

Mature adipocytes (day 6) were washed with PBS and fixed in 10% formaldehyde (#061–00416, Wako) for 10 min. The cells were washed with PBS and rinsed with 60% isopropanol (#166–04836, Wako) followed by staining with freshly prepared oil red O solution (O0625, Sigma-Aldrich) in 60% isopropanol. After the cells were rinsed with 60% isopropanol and PBS, images were scanned by a scanner. Representative oil red O staining data are shown, and more than two additional experimental replicates showed similar results.

## Animal study

Male C57BL/6J mice bred in our facility were used in all experiments. Global ASK1KO mice [39], ASK1$^{Flox/Flox}$; Adipoq-Cre/+ mice [9], and ASK1$^{Flox/Flox}$; LysM-Cre/+ mice [40] were described previously. A normal diet (NMF) was purchased from Oriental Yeast, and a high-fat diet (High Fat Diet 32) was purchased from CLEA Japan. WT mice and global ASK1KO mice were bred separately; however, littermate control mice were used for ASK1$^{Flox/Flox}$; Adipoq-Cre/+ mice and ASK1$^{Flox/Flox}$; LysM-Cre/+ mice. Mice were provided with a high-fat diet for 10 weeks, beginning at the age of 16 weeks. Glucose (1.0 g/kg weight) was injected intraperitoneally after an overnight fast for the glucose tolerance test, and 1 U/kg weight of insulin was injected intraperitoneally after 6 h of fasting for the insulin tolerance test. D (+)-glucose (#041–00595) was purchased from Wako, and insulin (Humulin R 100 U/mL) was purchased from Eli Lilly; both were dissolved in PBS. Blood glucose levels were monitored using MEDI-SAFE MINI (Terumo). All animal experiments were repeated on different days at least three times to minimize the effects of experimental conditions such as the breeding environment. All animal experiments were performed according to procedures approved by the Graduate School of Pharmaceutical Sciences, The University of Tokyo, and conformed to the guideline for the care and use of animals, published by The University of Tokyo.

## Peptide identification and quantification by mass spectrometry

HEK293T cells grown on 100-mm dishes were cotransfected with Flag-tagged ASK1 KR and PKA or empty vector using Lipofectamine 2000 (Invitrogen). Twenty-four hours after transfection, the cells were homogenized in lysis buffer (20 mM HEPES, pH 7.5, 150 mM NaCl, 50 mM NaF, 1 mM Na$_3$VO$_4$, 1% Triton-X100, 1 mM PMSF, 5 μg/ml leupeptin, 5 μg/ml aprotinin, and 3 μg/ml pepstatin A). The cell lysates were centrifuged at 20,000 x g for 10 min at 4°C, and the resultant supernatants were incubated for 1 h at 4°C with anti-Flag M2 beads (Sigma-Aldrich). The beads were washed three times, and immunoprecipitated proteins were eluted with 0.5 mg/ml Flag peptide.

After TCA precipitation, the immunoprecipitated protein pellet was resolubilized with 100 mM ammonium bicarbonate (AmBic, pH 8.8) containing 7 M guanidine hydrochloride and 0.01% decyl β-D-glucopyranoside (DG). The resolubilized proteins were reduced with 5 mM TCEP at 65˚C for 30 min and then alkylated with 10 mM iodoacetamide at 25˚C for 30 min. The alkylated samples were diluted 5-fold in 100 mM AmBic (pH 8.8) and 0.01% DG buffer containing 500 ng of lysyl endopeptidase. After being incubated at 37˚C for 3 h, the samples were diluted 2-fold in $H_2O$ containing 100 ng of trypsin and digested at 37˚C for 16 h. The resulting peptides were analyzed with a custom-made nanopump system (LC-Assist) coupled to a TripleTOF 5600+ spectrometer (Sciex). The samples were loaded on a custom-made C18 column (150 μm × 50 mm) packed with Mightysil RP-18 GP (3 μm) (Kanto Chemical). For elution, mobile phase A (H2O and 0.1% FA) and mobile phase B (0.1% FA in ACN) were used with an 80-min linear gradient from 0% to 40% B at a flow rate of 100 nL/min. Eluted peptides from the reversed-phase chromatography were directly loaded on the ESI source in positive ionization and high sensitivity modes. The MS survey spectrum was acquired in the range of 400 to 1500 m/z in 250 ms. For information-dependent acquisition (IDA), 25 precursor ions above a 50-count threshold with charge states +2 and +3 were selected for MS/MS scans. Each MS/MS experiment set the precursor m/z on a 12-s dynamic exclusion list, and the scan range was 100 to 1500 m/z over 100 ms. Peptide identification and precursor ion quantification were performed with ProteinPilot software (Sciex) using the NCBI nonredundant human protein data set (NCBI-nr RefSeq Release 71, containing 179460 entries). The tolerances were specified as ± 0.05 Da for peptides and ± 0.05 Da for MS/MS fragments [41].

## Statistics

All statistical analyses were performed using Microsoft Excel or Prism 8, and the details are described in the figure legends. Briefly, one sample t-test, unpaired two-tailed Welch's t-test or paired two-tailed t-test were used to compare two groups, the mixed-effects model (REML) followed by Dunnett's multiple comparisons test was used in Fig 3D, and q-values (false discovery rate (FDR)-adjusted p-values) for GTT and ITT were calculated using the Fisher LSD method followed by the two-stage step-up method of Benjamini, Krieger, and Yekutieli. The Mann-Whitney test followed by the two-stage step-up method of Benjamini, Krieger, and Yekutieli was used in S3E and S3F Fig, and the Mann-Whitney test alone was used in S3G Fig because the value of blood glucose reached the upper limit of detection (600 mg/mL) in 2 mice.

## Results

### The PKA-ASK1-p38/JNK pathway is activated in mature brown adipocytes

In our previous work, we demonstrated that the βAR-PKA-ASK1-p38 pathway is activated in immature brown adipocytes and contributes to brown adipocyte maturation [9]. No evidence has shown that the newly discovered signaling axis is retained in mature adipocytes, although multiple studies have shown that p38 is activated by βAR signaling [16–18] and cardiac natriuretic peptide [42] in adipocytes. We, therefore, treated mature brown adipocytes with a wide range of PKA activators, i.e., CL316,243 (a β3AR-specific agonist), isoproterenol (a pan-βAR agonist), forskolin (an adenylyl cyclase activator), norepinephrine (a pan-AR agonist) and 8-pCPT-cAMP (a cAMP analog), and monitored the activation status of kinases using phospho-specific antibodies. Similar to the results in immature brown adipocytes [9], we observed the activation of ASK and p38 (Fig 1A, S1A Fig). Additionally, all PKA activators induced the activation of JNK, the other pivotal player of downstream stress-responsive MAPK (Fig 1A). However, we cannot perfectly define which endogenous ASK proteins are activated by using

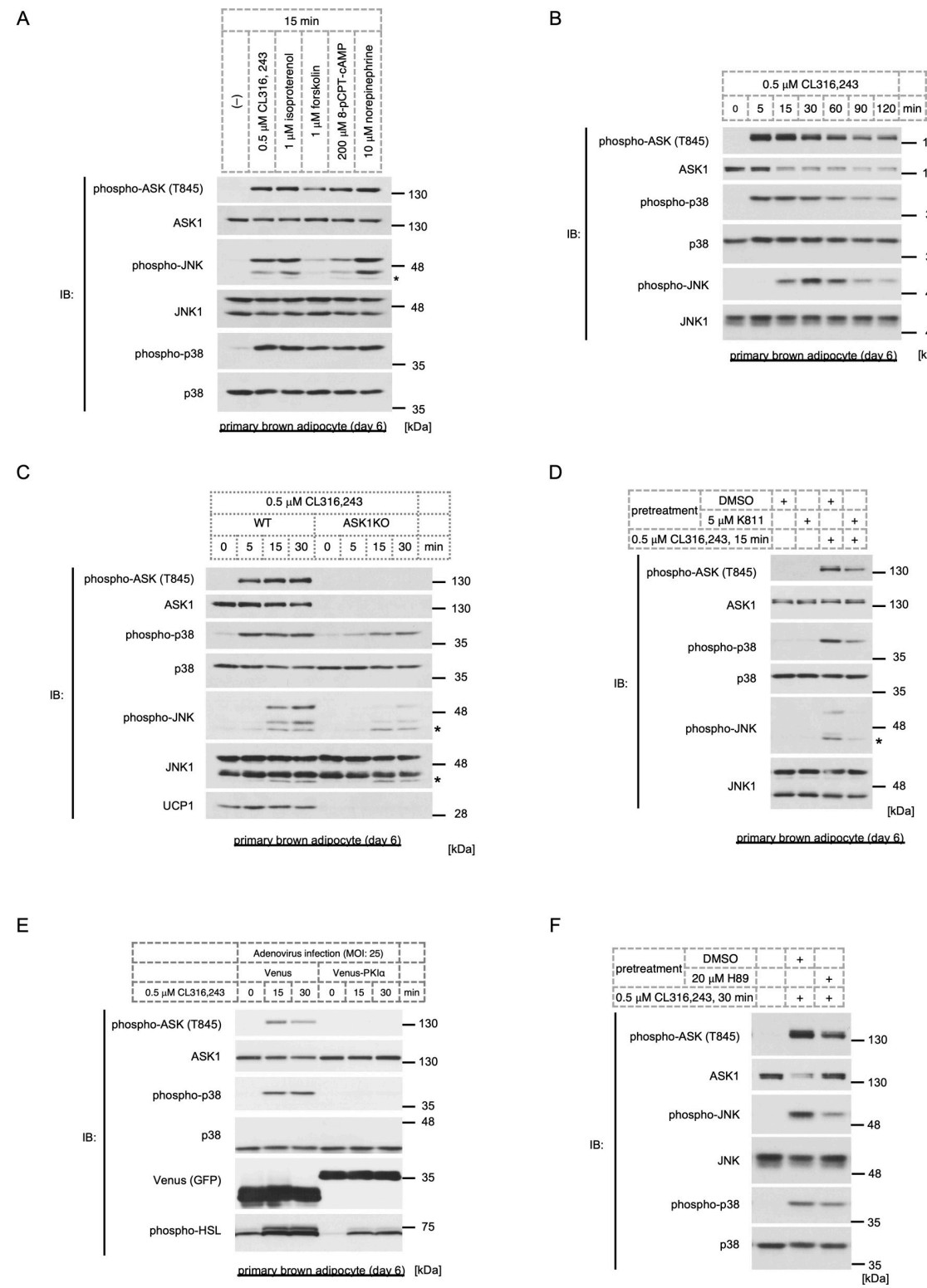

**Fig 1. The PKA-ASK1-p38/JNK pathway is activated in mature brown adipocytes.** (A) Western blot analysis of ASK and MAPK activation status in response to PKA activators (0.5 μM CL316,243 (a β3AR-specific agonist), 1 μM isoproterenol (a pan-βAR agonist), 1 μM forskolin (an adenylyl cyclase activator), 200 μM 8-pCPT-cAMP (a cAMP analogue) and 10 μM norepinephrine (a pan-AR agonist)) for 15 min in mature brown adipocytes (day 6). (B) The transition of ASK, p38, and JNK activity for the indicated

time in response to 0.5 μM CL316,243 in mature brown adipocytes (day 6), as assessed by Western blotting. (C) CL316,243-dependent ASK and MAPK activation and UCP1 expression in mature brown adipocytes (day 6) derived from wild-type or ASK1KO mice, as assessed by Western blotting. (D) Mature brown adipocytes (day 6) were pretreated with 5 μM K811, an ASK1 inhibitor, for 30 min and then treated with 0.5 μM CL316,243 for 15 min. ASK and MAPK activity was assessed by Western blotting. (E) Venus or Venus-tagged PKIα were overexpressed by adenovirus infection (MOI of 25), and CL316,243-dependent ASK and MAPK activation and the phosphorylation of HSL in mature brown adipocytes (day 6) were monitored by Western blotting. Venus was detected using an anti-GFP antibody. (F) Mature brown adipocytes (day 6) were pretreated with 20 μM H89, a PKA inhibitor, for 30 min and then treated with 0.5 μM CL316,243 for 30 min. ASK and MAPK activity was assessed by Western blotting.

an anti-phospho-ASK antibody because it recognizes all ASKs and because the molecular weights of ASKs are almost equivalent. However, the fact that (1) the major phospho-ASK band corresponded to the ASK1 band, (2) the major phospho-ASK band did not correspond to the ASK2 band (ASK2 is slightly smaller than ASK1), and (3) ASK3 was not detected either by Western blotting and qRT-PCR, suggests that endogenous ASK1 is predominantly activated in mature brown adipocytes. Both ASK and p38 rapidly reached the maximum activated state within 5 to 15 min after $\beta_3$AR activation, and the activity was sustained for at least 2 h, while JNK was slowly activated following ASK and p38 activation (Fig 1B). To assess the involvement of ASK1 in the activation of the downstream MAPKs, p38 and JNK, we prepared ASK1 knockout (ASK1KO) brown adipocytes. Consistent with our previous results [9], UCP1 expression was attenuated in mature ASK1KO brown adipocytes, while the abundance of MAPKs was comparable between the two genotypes. The results showed that CL316,243-dependent p38 and JNK activation were suppressed in ASK1KO cells, suggesting that ASK1 is an upstream kinase of p38 and JNK (Fig 1C). Because p38 and JNK activation were still observed in ASK1KO cells, we also analyzed ASK1, ASK2, and ASK3 triple knockout (ASKTKO) adipocytes, which lack all ASK family proteins and can normally differentiate (S1B–S1E Fig). However, p38 and JNK were still activated in ASKTKO cells (S1F Fig), suggesting the involvement of other MAP3Ks. To eliminate the effect of ASK1 deficiency on the development of adipocyte precursors *in vivo*, we inhibited the activity of ASK1 by administering a potent ASK1-specific inhibitor, K811 [43], after inducing differentiation. Since ASK1 activation is highly dependent on its autophosphorylation [34], the K811 effect was verified by the suppression of ASK1 phosphorylation. Consistent with the results in ASK1KO cells, both p38 and JNK activity were attenuated by K811, confirming that p38 and JNK are downstream targets of the ASK1 pathway (Fig 1D). Last, we used an adenovirus-mediated PKIα overexpression system to inhibit PKA action, and the effect of PKIα was confirmed by the suppressed phosphorylation of hormone-sensitive lipase (HSL) (Fig 1E), which is a known PKA target [44]. The data clearly showed that both ASK and p38 activity were diminished, suggesting that PKA serves as an upstream activator of the ASK-p38 axis (Fig 1E). Additionally, pretreatment with the PKA inhibitor H89 resulted in the apparent suppression of ASK and JNK activity with limited suppressive effects on p38 (Fig 1F). These data suggest that the PKA-ASK1-MAPK axis is activated by βAR signaling in mature brown adipocytes.

## The C-terminus of ASK1 is critical for PKA-dependent activation

Although PKA activates ASK1 in both immature/mature brown adipocytes (Fig 1E) [9] and mature white adipocytes [9], the mechanism remained unknown. Mouse ASK1 consists of 1380 amino acids with a kinase domain (KD) at the center flanked by N- and C-terminal regulatory regions (NT and CT, respectively) (Fig 2A). Accumulating evidence has shown that ASK1 activity is widely regulated by posttranslational modifications and protein-protein interactions at various regions in ASK1 [22, 23]; hence, we sought to determine the interactive region between ASK1 and PKA. However, the results of immunoprecipitation analysis

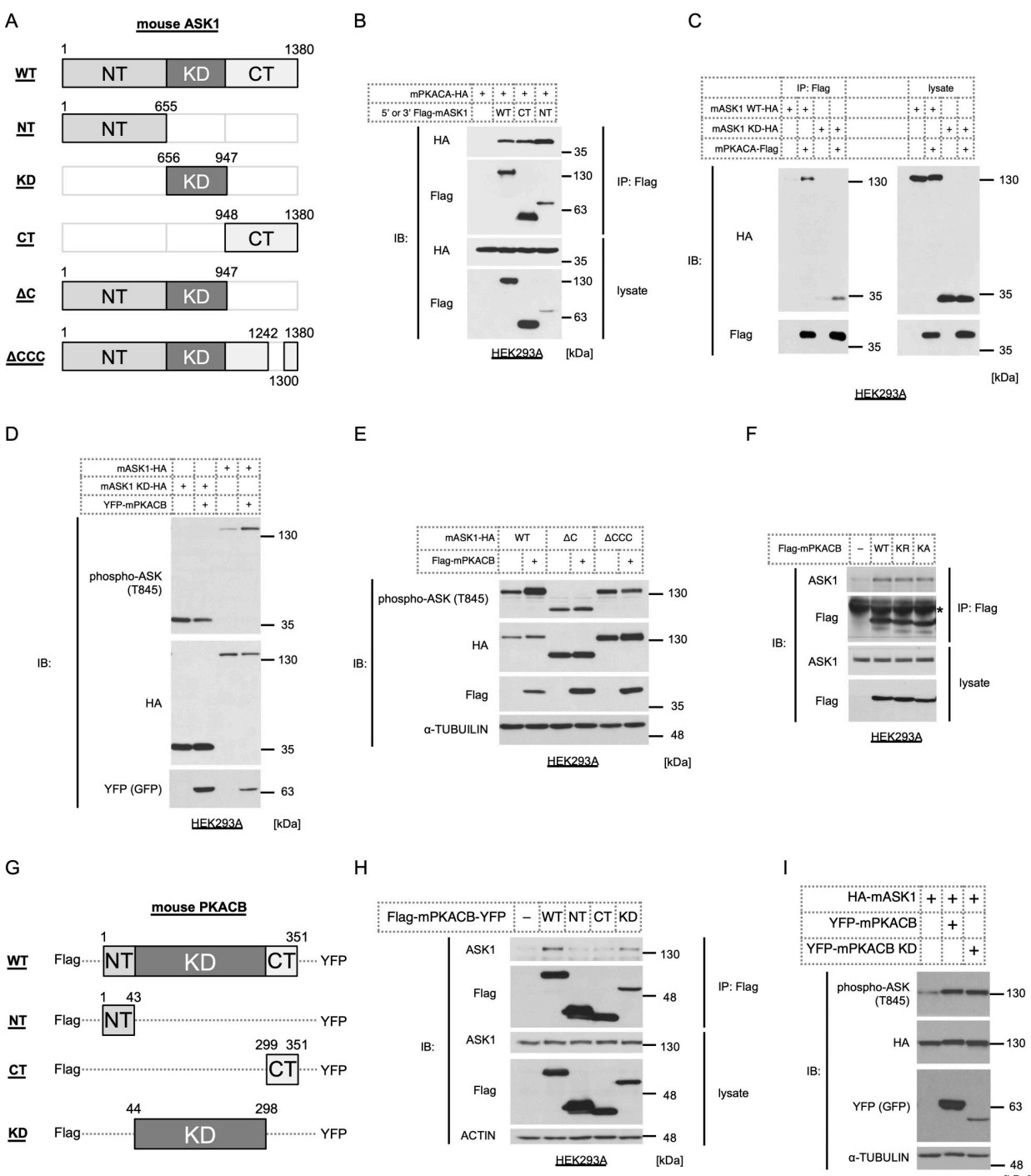

**Fig 2. C-terminus of ASK1 is critical for PKA-dependent activation.** (A) Schematics of WT and mutant mouse ASK1. The amino acid numbers are indicated. WT: wild-type, NT: N-terminus, KD: kinase domain, CT: C-terminus, ⊿C: delta C-terminus, ⊿CCC: delta C-terminus coiled-coil domain. (B) HEK293A cells were cotransfected with 3'-Flag-tagged WT mouse ASK1, 5'-Flag-tagged mouse ASK1 NT or 5'-Flag-tagged mouse ASK1 CT and 3-'HA-tagged mouse PKA catalytic subunit α (mPKACA) in and immunoprecipitated with anti-Flag antibody. The binding between mutant ASK1 and mPKACA was detected by Western blotting. (C) Coimmunoprecipitation assay between mouse WT ASK1 or the KD of ASK1 and mPKACA in HEK293A cells, similar to Fig 2B. (D, E) Western blotting for phosphorylation levels of the KD (D), ⊿C (E), and ⊿CCC (E) of ASK1 coexpressed with mouse PKA catalytic subunit β (mPKACB) in HEK293A cells. (F) Coimmunoprecipitation assay between exogenously expressed Flag-tagged WT or kinase-negative mutant mPKACB (K72R or K72A) and endogenous ASK1 in HEK293A cells. (G) Schematics of WT and mutant mouse PKACB. (H) Coimmunoprecipitation assay between exogenously expressed 5'-Flag-/3'-YFP-tagged WT or mutant mPKACB and endogenous ASK1 in HEK293A cells. (I) Western blotting for the phosphorylation levels of WT ASK1 coexpressed with WT or mutant mPKACB in HEK293A cells.

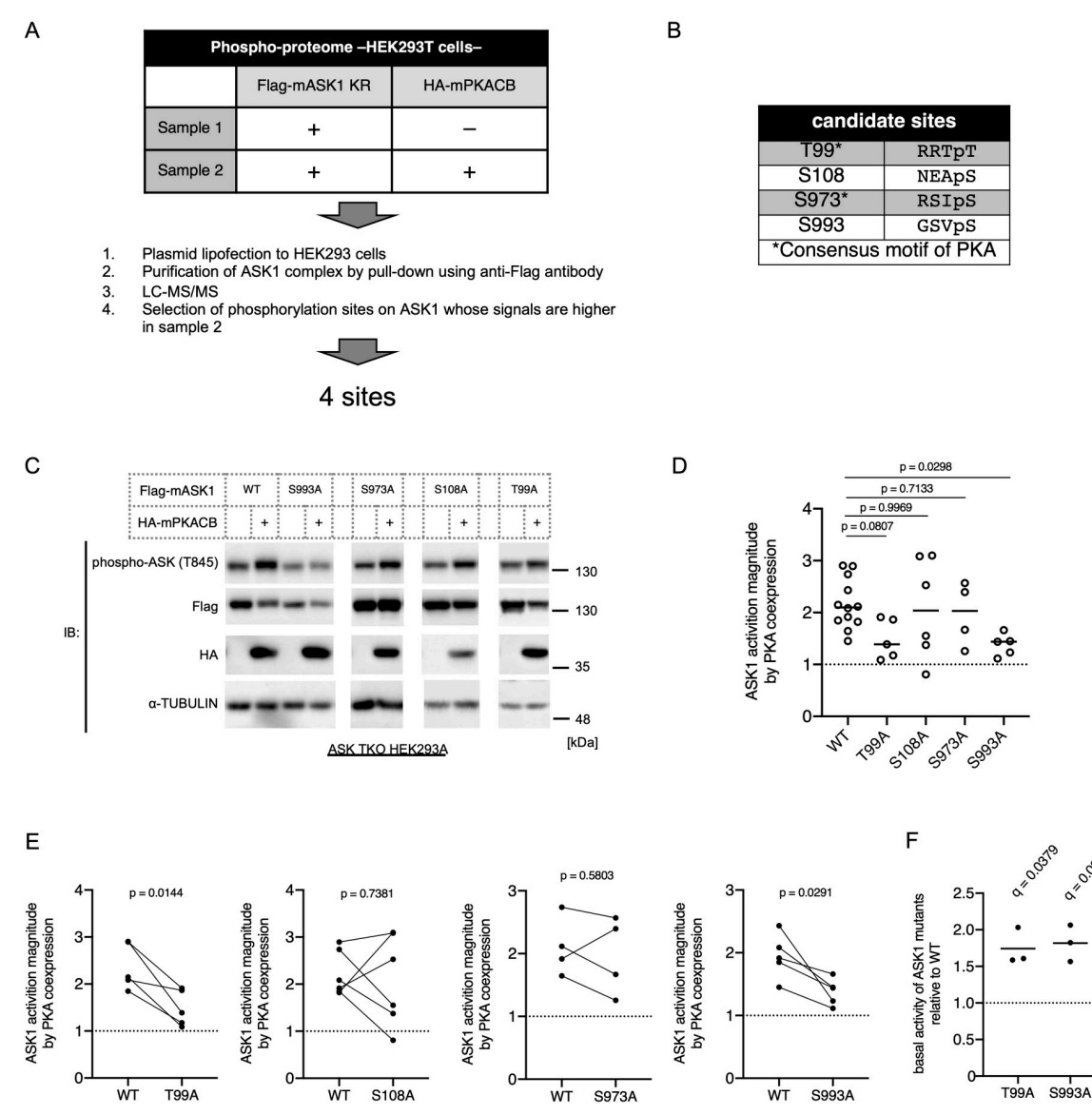

**Fig 3. PKA activates ASK1 through the phosphorylation of Ser 993.** (A) A schematic of the mass spectrometry analysis. (B) A list of PKA-dependent phosphorylation candidate sites on mouse ASK1. (C) Western blotting for the phosphorylation levels of WT and mutant ASK1 coexpressed with WT mPKACB in ASK1, ASK2, and ASK3 triple knockout HEK293A cells. The blots of each mutant are from different membranes. (D) Activation levels of WT and mutant ASK1 in the presence of mPKACB, calculated by the quantitative densitometric data from Western blots. Data were normalized to the activation levels of solely expressed WT or mutant ASK1 in each experiment. The means and individual data are shown. The mixed-effects model (REML) followed by Dunnett's multiple comparisons test was used. (E) The different ways of the data representation of Fig 3D. The individual data is shown, and the connecting lines indicate matched data in one experiment. A two-tailed paired t-test was used. (F) Basal activation levels of ASK1 mutants relative to WT. The means and individual data are shown. q values (FDR-adjusted p-values) were calculated using one sample t-test followed by the two-stage step-up method of Benjamini, Krieger, and Yekutieli.

revealed that PKA interacts with all of the ASK1 fragments tested, including the CT, KD, and NT (Fig 2B and 2C). From a different viewpoint, we examined the region of ASK1 required for PKA-dependent activation by partially depleting the regulatory region of ASK1 (Fig 2A). First, we coexpressed the KD of ASK1 with a PKA catalytic subunit, but the KD failed to be activated, whereas full-length ASK1 was activated, suggesting that the N-terminus and/or C-terminus are required for activation (Fig 2D). Next, we focused on the C-terminal region, which

harbors a predicted C-terminal coiled-coil (CCC) domain [34] and a sterile-alpha motif (SAM) domain [28], which is essential for high-order oligomerization of ASK1. Interestingly, exogenously expressed PKA did not activate both ⊿C and ⊿CCC of ASK1 (Fig 2E), which indicates that the C-terminal regulatory region, specifically a domain for homo-oligomerization, is a prerequisite for PKA-dependent ASK1 activation. Additionally, we examined the interaction between the two kinases by modulating PKA. Although we reported that kinase-negative forms of PKA, KR and KA mutants, failed to activate ASK1 [9], whether ASK1 binding is missing in those PKA mutants remained unclear. The coimmunoprecipitation analysis revealed that kinase-negative mutants of PKAs still bind ASK1 (Fig 2F), suggesting that kinase activity itself is vital for ASK1 activation. To further analyze the mechanism of action, we prepared truncated mutants of PKA. The PKA holoenzyme is composed of two catalytic subunits and two regulatory subunits; hence, the catalytic subunits contain only a few dozen amino acids on either side of the kinase domain (KD) (Fig 2G) [45]. However, multiple posttranslational modifications in the short flanking regions, including phosphorylation and myristoylation, contribute to the proper function of PKA [46, 47]. We analyzed the binding ability of each PKA fragment by coimmunoprecipitation and found that only KD retained its affinity for endogenous ASK1 (Fig 2H). Moreover, we confirmed that the KD of PKA retained the ability to activate ASK1 (Fig 2I). These results suggest that the KD of PKA interacts with ASK1 presumably at multiple sites and enhances ASK1 kinase activity through a C-terminus-dependent autophosphorylation mechanism. Since the kinase activity of PKA is a prerequisite for ASK1 activation, PKA functions by phosphorylating components of the ASK1 complex and/or ASK1 itself.

To further examine the precise mechanisms of ASK1 activation, we performed phosphoproteome analyses to discover phosphorylated sites on ASK1 evoked by PKA-kinase activity. Briefly, Flag-tagged ASK1 was exogenously expressed, and PKA-activated cells were mimicked by overexpressing PKA (Fig 3A). ASK1 was purified with an anti-Flag antibody, and phosphorylation sites were identified by LC-MS/MS (Fig 3A). A total of 77.3% of the ASK1 sequence was detected, and 13 potential phosphorylation sites were identified (S2A Fig). We selected the sites whose phosphorylation levels were augmented by the presence of PKA and then identified four candidate sites, namely, Thr 99, Ser 108, Ser 973, and Ser 993 (Fig 3B). It is noteworthy that the sequences adjacent to Thr 99 and Ser 973 match the consensus motif of PKA [48], suggesting that Thr 99 and/or Ser 973 are directly phosphorylated by PKA. Since Ser 973 (corresponding to human Ser 967) has already been identified as a 14-3-3 binding site [49], we used a commercially available antibody against phospho-ASK1 Ser 973. Although the phosphorylation of Ser 973 is usually inversely correlated with ASK1 activity [49, 50], five different PKA activators all induced phosphorylation at Ser 973 in parallel with activating ASK1 (S2B Fig). Additionally, phosphorylation at Ser 973 was well correlated with phosphorylation at Thr 845 at each time point after stimulation with CL316,243 (S1F Fig). Hence, we investigated whether phosphorylation at Ser 973 is required for PKA-dependent ASK1 activation. Both wild-type ASK1 and the ASK1 S973A were overexpressed in brown adipocytes derived from ASKTKO mice [51] to avoid the effects of endogenous ASKs because the ASK family can autophosphorylate each other to facilitate the activity. The results clearly showed that the S973A mutant was activated by CL316,243, indicating that phosphorylation at Ser 973 is not required for PKA-dependent ASK1 activation (S2C Fig). Therefore, we coexpressed all four mutants with PKA in the newly established ASKTKO HEK293A cells (S2D Fig) to assess the importance of each candidate phosphorylation site for ASK1 activation by mass spectrometry. Interestingly, both T99A and S993A mutants failed to be fully activated by PKA (Fig 3C–3E). Notably, the basal activity of the T99A and S993A mutants was significantly higher than that of WT (Fig 3F), which may limit the fold increase of ASK1 activity by PKA coexpression.

These results suggest that PKA directly or indirectly leads to phosphorylation at Thr 99 and/or Ser 993 and facilitates ASK1 activation. The fact that Ser 993 is located in the C-terminal region also supports the importance of the site (Fig 2A).

## PKA also activates ASK2

In the previous analysis, we discovered that ASK1 forms a high molecular mass signaling complex (>MDa, compared to ~150 kDa for the monomer) [52], and subsequent studies have shed light on a wide variety of components of this complex [27, 28], including ASK2, a member of the ASK family. ASK2 tightly forms a heteromeric complex with ASK1 and supports ASK1 kinase activity [25]. Considering the possibility that PKA indirectly activates ASK1, ASK2 is a potential mediator of the mechanism. Hence, we coexpressed ASK2 with PKA to examine the hypothesis. A kinase-negative mutant of ASK1 was also coexpressed because the expression of ASK2 alone is unstable and because ASK1 can prevent degradation [25]; we confirmed reduced ASK2 expression in ASK1 KO adipocytes (S2E Fig). The phospho-specific antibody against mouse ASK1 T845 can recognize phospho-T807 on mouse ASK2 as well; hence, we can monitor the activation status of both kinases solely by this antibody [25]. Both a kinase-negative mutant of ASK1 and ASK2 were phosphorylated by the overexpression of PKA, suggesting that PKA activates ASK2 and, subsequently, ASK2 phosphorylates the ASK1 mutant (Fig 4A). This result raises the possibility that PKA potentially can activate both ASK1 and ASK2 and that the ASK1-ASK2 complex is activated synergistically; hence, we analyzed whether ASK1 and ASK2 are solely activated by PKA using mature ASKTKO brown adipocytes. ASK1 or ASK2 was exogenously expressed by the adenovirus-mediated system, and the cells were stimulated by CL316,243. Although activation levels were suppressed compared to those in WT cells, both ASK1 and ASK2 were activated upon CL316,243 treatment in ASKTKO cells (Fig 4B and 4C), which suggests that the PKA axis can activate both ASK1 and ASK2. To examine the involvement of ASK2 in MAPK signaling in mature brown adipocytes, ASK2 knockout (ASK2KO) adipocytes were stimulated with CL316,243. The results showed that both p38 and JNK were suppressed (Fig 4D), suggesting that ASK2 also contributes to MAPK signaling in mature brown adipocytes. However, ASK1 levels were also attenuated by ASK2 deficiency, which may contribute to the suppression of MAPK activity. Moreover, the fact that PKA activates ASK2 in mature adipocytes suggests that the PKA-ASK2 axis may function even in immature adipocytes and contribute to differentiation-dependent gene regulation, similar to ASK1 [9]. Oil red O staining showed that ASK2 was not necessary for lipid accumulation in brown adipocytes (Fig 4E); however, UCP1 expression was dramatically reduced in ASK2KO adipocytes (Fig 4D), similar to ASK1KO adipocytes [9].

## Global knockout of ASK1 aggravates glucose clearance in high-fat diet-fed mice

Our previous work discovered ASK1 regulates oxygen consumption in brown adipose tissue through brown adipocyte maturation [9]. Additionally, we found that white adipose tissue weight ratio was significantly higher in high-fat diet (HFD)-fed ASK1KO mice [9], suggesting the involvement of ASK1 in the physiology of obesity. However, we did not analyze glucose metabolism, which is one of the most prominent outcomes of obesity. Multiple rodent studies have shown that the depletion of ASK1 action throughout the body or selectively in adipose tissues leads to the amelioration of HFD-induced insulin resistance [53–55]; however, one human study suggests that ASK1 expression in skeletal muscle positively correlates with *in vivo* insulin action [56]. Additionally, the increased adipose tissue weight in HFD-fed ASK1KO mice has not been described elsewhere. Therefore, we attempted to confirm the

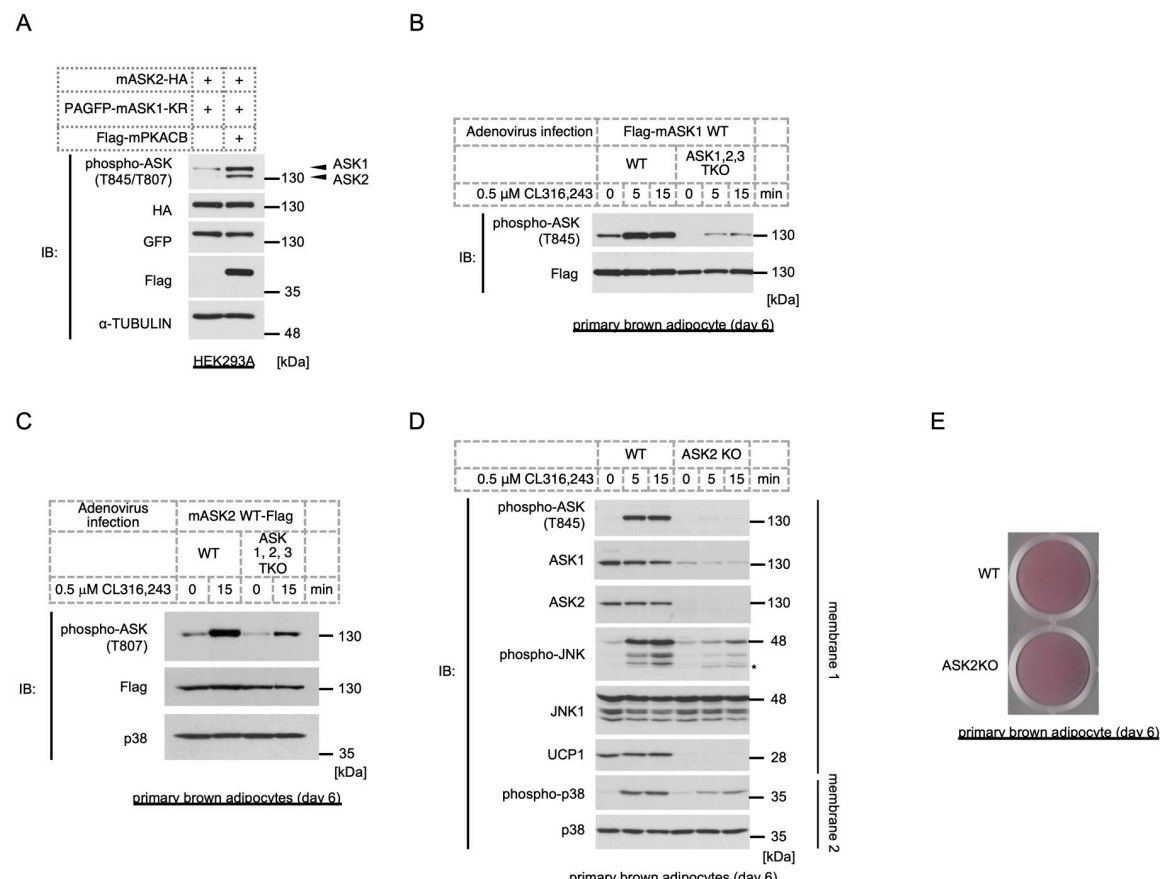

**Fig 4. ASK2 is also activated by PKA and regulates MAPK signaling in concert with ASK1.** (A) Western blotting analysis of ASK2 phosphorylation levels coexpressed with PKACB. A kinase-negative mutant of ASK1 (K716R) was also coexpressed to stabilize ASK2. (B, C) Wild-type mouse ASK1 (B) or ASK2 (C) was overexpressed by adenovirus infection in immature WT or ASKTKO cells (day 2), and differentiated mature brown adipocytes (day 6) were treated with CL316,243 for the indicated time. The phosphorylation of ASKs was assessed by Western blotting. (D) The activation of ASK and MAPKs in WT and ASK2KO mature brown adipocytes (day 6) in response to CL316,243 for the indicated time. (E) Oil red O staining of WT and ASK2KO mature brown adipocytes (day 6).

involvement of ASK1 in glucose clearance and insulin sensitivity by analyzing global ASK1KO mice. First, normal diet-fed mice were subjected to a glucose tolerance test (GTT) and an insulin tolerance test (ITT) after an overnight fast or 6 h of fasting, respectively. The results showed that there was no difference in glucose clearance or insulin sensitivity between the two genotypes (S3A–S3C Fig), which is consistent with a previous study, as evidenced by hyperglycemic clamp and hyperinsulinemic-euglycemic clamp analyses [57]. However, we found that glucose clearance was significantly aggravated in global ASK1KO mice after 10 weeks of HFD exposure (S3D–S3G Fig). In the ITT, blood glucose levels were significantly higher in global ASK1KO mice at each time point (S3D and S3H Fig), but the transition of the blood glucose ratio was comparable (S3I Fig). These results suggest that ASK1 is not involved in insulin responses. The body weight of ASK1KO mice before and after HFD exposure was slightly lower (S3A and S3J Fig). Notably, the area under the curve (AUC) of the GTT results was higher, particularly in the high body weight group of ASK1KO mice (>40 g weight) (S3K Fig), suggesting that the effect of ASK1 deficiency is more robust in severely obese animals. These results provide evidence that ASK1 has an accelerating function in glucose clearance in an HFD-induced obesity model.

We next asked which organs are responsible for the phenotype. Since adipocytes and inflammatory cells are one of the major players that are involved in glucose homeostasis, we utilized adiponectin-Cre to target adipocytes and LysM-Cre to target the myeloid lineage for the tissue-specific depletion of ASK1. Unfortunately, neither tissue-specific ASK1KO mouse exhibited phenotypes of global ASK1KO mice (S3L–S3Q Fig), suggesting that ASK1 in white/ brown adipocytes and myeloid cells, such as macrophages and monocytes, is not involved in glucose clearance.

## Discussion

Our current dataset shows that the βAR-PKA-ASK1-p38 signaling pathway is activated in mature brown adipocytes in addition to immature brown adipocytes [9]. Additionally, JNK was discovered to be an additional component of the βAR signaling cascade in mature brown adipocytes; CL316,243-dependent JNK activation in immature adipocytes (day 2) was not observed, though JNK1 was abundantly expressed. One report showed that a transcription factor, c-JUN, which is regulated by JNK activity, counteracts PKA-CREB pathway-dependent *Ucp1* induction in precursor cells [58]; hence, some JNK-inhibitory mechanisms may exist only in immature adipocytes to facilitate *Ucp1* expression. In mature brown adipocytes, since UCP1 is already abundantly expressed, JNK may function as a *Ucp1*-suppressive system to maintain an adequate amount of UCP1. The fact that the adipocyte-specific overexpression of dominant-negative JNK *in vivo* results in an increase in energy expenditure [59] suggests that the augmentation of UCP1 may be involved in this process.

Conversely, p38 is activated in both immature and mature brown adipocytes (Fig 1) [9]. Some pioneering works mainly executed by Dr. Collins laboratory have shown that βAR agonists evoke p38 activation in multiple adipocyte cell lines, differentiated brown adipocytes, and *in vivo* [16–19]. Mitogen-activated protein kinase kinase 3 (MKK3) may function as a MAP2K, connecting ASK1 and p38 [18]. Additionally, the requirement of p38 activity for *Ucp1* expression has been demonstrated in multiple ways [16–18]. In our previous study, we revealed that the PKA-ASK1-p38 axis in immature adipocytes contributes to a subset of gene expression, including *Ucp1* [9]. However, a recent report showed that the adipocyte-specific deletion of p38α has a limited effect on brown adipose tissue status, e.g., it has almost no effect on UCP1 expression [60]. This result suggests that p38α in mature brown adipocytes is not involved in UCP1 expression, but p38α may still contribute to UCP1 expression in immature adipocytes because it has not been confirmed that p38α is absent in immature adipocytes in their model. Several factors bring the controversy surrounding p38 functions. In essence, p38 family consists of p38α, β, γ, and δ, widely used inhibitors inhibit only α and β, and compensatory mechanisms in the absence of one of the members are mostly unknown. Hence, further precise analyses are required to conclude the role of the p38 family in brown adipocytes.

We identified the phosphorylation of ASK1 at Thr 99 and Ser 993 as potentially critical modifications for PKA-dependent activation (Fig 3). The sequences adjacent to mouse Thr 99 and Ser 993 are conserved in human and rat ASK1, and the phosphorylation of equivalent serines to mouse Ser 993 was confirmed by proteomics elsewhere [61–63]. Importantly, Thr 99 is a consensus target of PKA (Fig 3B) [48], raising the possibility PKA can directly phosphorylate this threonine. Although we identified phosphorylated peptides by mass spectrometry, PKA-dependent phosphorylation should be validated in future studies using phospho-specific antibodies because experiments using SA mutants cannot exclude the possibility of the involvement of other types of posttranslational modifications, such as glycosylation. Additionally, an *in vitro* kinase assay is required to conclude whether the phosphorylation is direct or indirect. Ser 993 is retained in the ⊿CCC of ASK1, but the ⊿CCC loses its ability to be activated by

PKA (Fig 2E), which suggests that both phospho-ASK1 Ser 993 and CCC-mediated homo-oligomerization are required for its activation; however, the mutual interaction of these two phenomena is still unknown. A recent study suggested that the SAM domain is located on the C-terminal side of the CCC and is critical for oligomerization [28]; hence, the effects of S993A and the ⊿CCC on the SAM domain structure should also be considered. Notably, we could not find similar sequences around mouse ASK1 Thr 99 or Ser 993 in mouse ASK2, suggesting that PKA differentially regulates ASK2.

The data obtained in ASKTKO adipocytes clearly showed that both ASK1 and ASK2 could be activated by PKA signaling (Fig 4B and 4C); however, ASK1KO [9], ASK2KO (Fig 4D), and ASKTKO (S1F Fig) all partially suppressed CL316,243-induced p38 and JNK activation. Since the level of ASK1 was dramatically reduced in ASK2KO cells (Fig 4D) and ASK2 protein is degraded in the absence of ASK1 [25], we cannot conclude which ASK kinase is mainly responsible for the activation of MAPKs in mature brown adipocytes. Consistent with the notion that ASK3 is selectively expressed in the kidney [35], we did not detect ASK3 in mature brown adipocytes either by Western blotting or qRT-PCR; hence, we have no evidence of the involvement of ASK3 for now.

Consistent with our previous study [9], the global knockout of ASK1 did not affect HFD-dependent body weight gain (S3D and S3J Fig). However, glucose clearance capacity was significantly aggravated by ASK1 deficiency, suggesting a potential role for ASK1 in glucose homeostasis. Previous studies from other laboratories have shown that ASK1 has inhibitory effects on glucose clearance [53–55]; mouse strain, age of the mouse, the type of HFD, and the rearing environment may explain the discrepancy. It is noteworthy that glucose clearance ability is markedly impaired, especially in severely obese ASK1KO mice, suggesting that ASK1 differentially functions based on the level of obesity (S3K Fig).

Subsequent experiments using tissue-specific knockout models revealed that neither adipocytes nor myeloid cells are responsible for the phenotype (S3L–S3Q Fig). A recently published study revealed that the hepatocyte-specific knockout of ASK1 in HFD-fed mice leads to a higher degree of hepatic steatosis, inflammation, and fibrosis, along with slightly impaired glucose clearance [64]. Therefore, hepatic fibrosis induced by ASK1 depletion in hepatocytes may at least partially explain the mechanism. Nonetheless, we observed a significant increase in fasting glucose levels in global ASK1KO mice (S3E and S3H Fig), but fasting glucose levels did not seem to be altered in hepatocyte-specific ASK1KO mice [64], suggesting that there are other ASK1-dependent regulatory mechanisms for glucose homeostasis. One possibility is a malfunction in insulin secretion, but this is unlikely given the previous results using the hyperglycemic clamp method [57]; however, impaired insulin secretion in fasted HFD-fed ASK1KO mice is possible.

We believe our findings provide novel insight into the regulation of mature brown adipocytes, which may be beneficial for developing βAR-dependent strategies against energy metabolism-related diseases.

## Supporting information

**S1 Table. Primer sequences.**
(DOCX)

**S1 Fig. CL316,243-dependent p38 and JNK activation is attenuated in ASKTKO brown adipocytes.** A. Brown adipocytes imaged at day 2 (immature) and day 6 (mature). B-D. Oil red O staining (B), Western blot analysis (C), and qRT-PCR analysis (D) of WT and ASKTKO mature brown adipocytes (day 6). The means and individual data are shown (N = 5) in D. E. 10 nM insulin was treated 30 min before measuring glucose uptake in mature brown

adipocytes (day 6). The means and individual data are shown (N = 3). Two-stage linear step-up procedure of Benjamini, Krieger and Yekutieli was used to adjust p-values. F. CL316,243-dependent MAPK activation in mature brown adipocytes (day 6) derived from wild-type and ASKTKO mice, as assessed by Western blotting.
(TIF)

**S2 Fig. Phosphorylation of ASK1 at Ser 973 is induced by CL316,243 but is not required for ASK1 activation.** A. The amino acid sequence of mouse ASK1 used in mass spectrometry analysis. Upper case: identified amino acids, lower case: unidentified amino acids, magenta (bold font): identified phosphorylation sites, magenta/bold/italicized (with an amino acid number): sites in which phosphorylation levels are augmented by PKA overexpression. B. Western blot analysis of ASK1 phosphorylation at Ser 973 in response to treatment with a PKA activator for 15 min in mature brown adipocytes (day 6). C. 3'Flag-tagged wild-type ASK1 or mutant ASK1 S973A was overexpressed by adenovirus infection in immature ASKTKO cells (day 2), and differentiated mature brown adipocytes (day 6) were treated with CL316,243 for the indicated time. The phosphorylation of ASK1 was assessed by Western blotting. D. Western blots for ASK1, ASK2, and ASK3 in control and ASKTKO monoclonal HEK293A cells. This is a representative of knockout cell screens. ASKTKO #5 was used in Fig 3C. E. Western blots for ASK2 expression in ASK1KO mature brown adipocytes (day 6).
(TIF)

**S3 Fig. Global knockout of ASK1 aggravates glucose clearance in high-fat diet-fed mice.** Blood glucose levels (mg/mL) from the glucose tolerance test (GTT) (B, E, F, M, P) and insulin tolerance test (ITT) (C, H, I, N, Q). The body weight of the mice assessed by the GTT and ITT are shown in A, D, L, and O, and unpaired two-tailed Welch's t-test was used. The body weight of global ASK1KO mice before the start of high-fat diet feeding was plotted and analyzed by unpaired two-tailed Welch's t-test (J). A to C are 28-week-old male WT (N = 8) and global ASK1KO mice (N = 8). D to K are 26-week-old high-fat diet-fed WT (N = 27 for GTT and N = 19 for ITT) and ASK1KO mice (N = 20 for GTT and N = 12 for ITT). L to N are 26-week-old high-fat diet-fed ASK1$^{Flox/Flox}$; +/+ (N = 7) and ASK1$^{Flox/Flox}$; LysM-Cre/+ mice (N = 8 for GTT and N = 7 for ITT). O to Q are 26-week-old high-fat diet-fed ASK1$^{Flox/Flox}$; +/+ (N = 7) and ASK1$^{Flox/Flox}$; Adipoq-Cre/+ mice (N = 11). The area under the curve (AUC) of S3E Fig was plotted and analyzed by the Mann-Whitney test (G). Correlation of body weight after 10 weeks of high-fat diet exposure and the AUC of the GTT (K). Two data points (filled triangles) represent the data from the mice in which the blood glucose level reached the upper limit of detection. Hence, linear regression curves are shown only for reference. Means ± 95% CI are shown for the GTT and ITT data. The means and individual data are shown for body weight and AUC data. q-values (FDR-adjusted p-values) for the GTT and ITT results were calculated using the Fisher LSD method followed by the two-stage step-up method of Benjamini, Krieger, and Yekutieli (for B, C, H, I, M, N, P, Q). Since the blood glucose level reached the upper limit of detection (600 mg/mL) in 2 mice, the Mann-Whitney test followed by the two-stage step-up method of Benjamini, Krieger, and Yekutieli was used in E and F. See R and S for actual q-values. *q < 0.05, **q < 0.01, ***q < 0.001.
(TIF)

**S1 Raw data.**
(XLSX)

**S1 Raw images.**
(PDF)

## Acknowledgments

We thank Kyowa Hakko Kirin for providing the ASK1 inhibitor K811. We also thank all the current and previous members of the Laboratory of Cell Signaling for meaningful discussion.

## Author Contributions

**Conceptualization:** Kazuki Hattori, Hidenori Ichijo.

**Data curation:** Kazuki Hattori.

**Formal analysis:** Kazuki Hattori, Hiroaki Wakatsuki, Tomohisa Hatta.

**Funding acquisition:** Kazuki Hattori, Hidenori Ichijo.

**Investigation:** Kazuki Hattori, Hiroaki Wakatsuki, Chihiro Sakauchi, Shotaro Furutani, Sho Sugawara, Tomohisa Hatta.

**Methodology:** Kazuki Hattori, Tomohisa Hatta, Tohru Natsume.

**Project administration:** Kazuki Hattori.

**Resources:** Tomohisa Hatta, Tohru Natsume, Hidenori Ichijo.

**Supervision:** Kazuki Hattori, Tohru Natsume, Hidenori Ichijo.

**Visualization:** Kazuki Hattori.

**Writing – original draft:** Kazuki Hattori, Hidenori Ichijo.

**Writing – review & editing:** Kazuki Hattori, Tomohisa Hatta, Hidenori Ichijo.

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
