## [Decision Letter · Decision Letter 0]

8 May 2020

PONE-D-20-11137

β-adrenergic receptor signaling evokes the PKA-ASK axis in mature brown adipocytes

PLOS ONE

Dear Dr. Hattori,

Thank you for submitting your manuscript to PLOS ONE. After careful consideration, we feel that it has merit but does not fully meet PLOS ONE’s publication criteria as it currently stands. Therefore, we invite you to submit a revised version of the manuscript that addresses the points raised during the review process.

Your manuscript was reviewed by two knowledgeable referees in this area and their comments are appended. As you will see while they find your work of interest, they both have raised several points that will need to be properly addressed by the authors before I can proceed further. In particular, Reviewer#1 raised issues regarding potential adverse effects of viral transfection on brown adipocyte differentiation as well as the traits of ASKKO cells. The authors need to address all their comments to fully satisfy both reviewers.

We would appreciate receiving your revised manuscript by Jun 22 2020 11:59PM. To enhance the reproducibility of your results, we recommend that if applicable you deposit your laboratory protocols in protocols.io, where a protocol can be assigned its own identifier (DOI) such that it can be cited independently in the future. For instructions see: http://journals.plos.org/plosone/s/submission-guidelines#loc-laboratory-protocols

We look forward to receiving your revised manuscript.

Kind regards,

Makoto Kanzaki, Ph.D.

Academic Editor

PLOS ONE

Journal Requirements:

Reviewers' comments:

Reviewer's Responses to Questions

**Comments to the Author**

1. Is the manuscript technically sound, and do the data support the conclusions?

Reviewer #1: Yes

Reviewer #2: Yes

2. Has the statistical analysis been performed appropriately and rigorously? 

Reviewer #1: Yes

Reviewer #2: Yes

3. Have the authors made all data underlying the findings in their manuscript fully available?

Reviewer #1: Yes

Reviewer #2: Yes

4. Is the manuscript presented in an intelligible fashion and written in standard English?

Reviewer #1: Yes

Reviewer #2: No

5. Review Comments to the Author

Reviewer #1: In the present manuscript, the authors identified the ASK family as a crucial signaling molecule bridging PKA and MAPK in mature brown adipocytes. Mechanistically, the phosphorylation of ASK1 at threonine 99 and serine 993 is critical in PKA-dependent ASK1 activation. Additionally, PKA also activates ASK2, which contributes to MAPK regulation. However, a couple of issues need to be addressed.

Major points:

1. In this study, immature brown adipocytes were treated with adenovirus-mediated PKIα, WT mouse ASK1, mouse ASK1 S973A, and mouse ASK2, on day 2. The authors should check the effect of the viral infection (PKIα, ASK1, S973A, and ASK2) on the differentiation of brown fat cells.

2. As the author mentioned, the expression of ASK2 alone is unstable and because ASK1 can prevent degradation. However, in Figure 1C, p38 and JNK activation could be observed in ASK1KO cells. The authors should check the expression of the other ASKs in ASK1KO cells.

3. In Figure 4B,C,D, the authors used the antibodies for phospho-ASK (T845) and phospho-ASK (T807) to indicate the activation of ASK, respectively. What is the difference between these two phosphorylation sites? The authors should add information about these two phosphorylation sites.

4. The authors should check the differentiation makers (aP2, PPARγ, c/EBPα, PGC1α) of ASKKO adipocytes by western blot and qPCR.

Reviewer #2: In this manuscript, authors further studied the function of ASK1 on beta3 adrenergic signaling pathway in mature brown adipocyte. Same with immature brown adipocyte, ASK1 was also required in this pathway. Besides, they found that the phosphorylation of Thr99 and Ser33 was critical for PKA-dependent AKS1 activation. The knockout of ASK1 or ASK2 both reduced the expression of UCP1 in mature brown adipocyte. These experiments were well designed and performed.

But it is not clear why authors did not continue to investigate the effect of ASK on thermogenesis in mature adipocyte but to examine its function in glucose clearance. It is confusing that why the specific deletion of ASK1 in adipose tissue had no effect on glucose clearance, since adipose tissue is also responsible to glucose absorbance.

1. ASK1 can upregulate the expression of Ucp1 in immature brown adipocyte, and it also promoted Ucp1’s expression in mature brown adipocyte. Is there any mechanic difference?

2. Blood glucose concentrations should be tested in ASK1 AKO mice after CL injection, since CL is helpful to clear glucose.

3. How about glucose uptake in ASK KO adipocyte in vitro?

4. Gene names need to be italic. All characters in a protein name need to be capitalized.

6. PLOS authors have the option to publish the peer review history of their article (what does this mean?). If published, this will include your full peer review and any attached files.

Reviewer #1: No

Reviewer #2: No

---

## [Author Response · Author response to Decision Letter 0]

24 Aug 2020

Response to Reviewers

Reviewer #1: In the present manuscript, the authors identified the ASK family as a crucial signaling molecule bridging PKA and MAPK in mature brown adipocytes. Mechanistically, the phosphorylation of ASK1 at threonine 99 and serine 993 is critical in PKA-dependent ASK1 activation. Additionally, PKA also activates ASK2, which contributes to MAPK regulation. However, a couple of issues need to be addressed.

Answer: 

Thank you for pointing out the critical issues. Please see below for the answers to your queries.

Major points:

1. In this study, immature brown adipocytes were treated with adenovirus-mediated PKIα, WT mouse ASK1, mouse ASK1 S973A, and mouse ASK2, on day 2. The authors should check the effect of the viral infection (PKIα, ASK1, S973A, and ASK2) on the differentiation of brown fat cells.

Answer:

Thank you for raising the issue. Though we do not have quantitative data in hand for differentiation levels of each sample, we always checked lipid droplet accumulation by microscopic analyses right before applying stimulation; adenoviral infection had no effect. Additionally, we want to emphasize that we overexpressed ASK1 WT, ASK1 S973A, or ASK2 for evaluating their responses to CL316,243, not for other endogenous signaling cascades. The results conclude that all of those exogenously expressed kinases can be activated. Even though the overexpression partially modulates adipocyte differentiation, the conclusion holds. Additionally, overexpression of PKIa provides supportive information along with the data using PKA inhibitor, H89 (Fig 1F); the results consistently show that PKA inhibition leads to the suppression of ASK-MAPK axis. Therefore, we believe that we have already included sufficient information for our conclusion.

2. As the author mentioned, the expression of ASK2 alone is unstable and because ASK1 can prevent degradation. However, in Figure 1C, p38 and JNK activation could be observed in ASK1KO cells. The authors should check the expression of the other ASKs in ASK1KO cells.

Answer:

Thank you for the question. As you mentioned, p38 and JNK are still activated in ASK1KO adipocytes; however, we observed their activation even in ASK1, 2, 3TKO cells, postulating the involvement of other MAP3Ks (please see S1F Fig and from line 286 of the results). Therefore, we concluded that ASK1 and/or ASK2 is a part of upstream signaling of p38 and JNK axes. Nonetheless, we agree that other ASK family expression in ASK1KO cells is of importance; hence, we have newly added WB data in S2E Fig, where we confirmed reduced expression of ASK2 in ASK1KO cells (please see line 430 as well). ASK3 was detected neither by WB nor qRT-PCR (please see from line 557 in the discussion).

3. In Figure 4B,C,D, the authors used the antibodies for phospho-ASK (T845) and phospho-ASK (T807) to indicate the activation of ASK, respectively. What is the difference between these two phosphorylation sites? The authors should add information about these two phosphorylation sites.

Answer:

We apologize for the confusion. We can monitor the activation of mouse ASK2 by assessing phosphorylation at T807, which can be detected by the anti-phospho-ASK (T845) antibody. The amino acid sequences around mouse ASK1 T845 and mouse ASK2 T807 are almost identical; hence, we can detect their phosphorylation by a single type of phospho-specific antibody (please see these articles for details: J Cell Physiol. 2002;191: 95–104 and J Biol Chem. 2007;282: 7522–7531). We have added information in the manuscript (please see from line 430 in the results).

4. The authors should check the differentiation makers (aP2, PPARγ, c/EBPα, PGC1α) of ASKKO adipocytes by western blot and qPCR.

Answer: 

Thank you for pointing out this crucial issue. We have performed Western blotting and qRT-PCR for comparing the expression status of WT and ASK1, 2, 3 TKO brown adipocytes and confirmed that a wide range of differentiation markers are not affected by the ASK family knockout (please see newly added S1C and D Figs).

Reviewer #2: In this manuscript, authors further studied the function of ASK1 on beta3 adrenergic signaling pathway in mature brown adipocyte. Same with immature brown adipocyte, ASK1 was also required in this pathway. Besides, they found that the phosphorylation of Thr99 and Ser33 was critical for PKA-dependent AKS1 activation. The knockout of ASK1 or ASK2 both reduced the expression of UCP1 in mature brown adipocyte. These experiments were well designed and performed.

But it is not clear why authors did not continue to investigate the effect of ASK on thermogenesis in mature adipocyte but to examine its function in glucose clearance. It is confusing that why the specific deletion of ASK1 in adipose tissue had no effect on glucose clearance, since adipose tissue is also responsible to glucose absorbance.

Answer: 

Thank you for your overall positive assessment of our work. As you noticed, we have already reported that ASK1 in adipocytes contributes to thermogenesis in brown adipocytes, at least partially through gene regulation, including Ucp1. Our lines of evidence suggest that ASK1-dependent gene regulation in immature adipocytes is critical for the thermogenic function of mature adipocytes. However, we have not denied the involvement of ASK1 in mature adipocytes for thermogenesis; it is technically challenging to distinguish ASK1 roles in two types of cells, especially in vivo. For instance, UCP1 expression is already diminished in ASK1KO mature brown adipocytes, leading to suppressed thermogenic ability in ASK1KO cells, which limits to examine the mature adipocyte-specific thermogenic function of ASK1. 

Therefore, we moved on to analyzing glucose metabolism as a different outcome, especially in obese conditions, exploiting our previous knowledge (please see from line 465 in the results). However, the results from the adipocyte-specific ASK1 knockout model clearly suggest that ASK1 in adipocytes has no impact on glucose homeostasis. One report from another group postulates the importance of hepatocyte-ASK1 for blood glucose homeostasis in obese mice; this may explain the reason why we did not observe impairment of glucose clearance in adipocyte-specific models (please see from line 568 in the discussion and EMBO Mol Med. 2019;0: e10124). Since these unexpected results are slightly off the main topic in the current manuscript, we placed them in the supplementary data and put much focus on ASK-pathway per se.

1. ASK1 can upregulate the expression of Ucp1 in immature brown adipocyte, and it also promoted Ucp1’s expression in mature brown adipocyte. Is there any mechanic difference?

Answer: 

Thank you for the question. We want to clarify that ASK1 in immature adipocytes contributes to Ucp1 expression throughout the differentiation; however, we do not have any pieces of evidence showing the similar function of ASK1 in mature adipocytes. βAR agonist can activate PKA-ASK-MAPK axes even in mature adipocytes, but the downstream events remain to be elucidated.

2. Blood glucose concentrations should be tested in ASK1 AKO mice after CL injection, since CL is helpful to clear glucose.

Answer: 

Thank you for the crucial suggestion. We fully agree that activated ASK1 in adipocytes might play a pivotal role in glucose clearance. However, our manuscript is digging more in the signaling cascade itself and put in vivo dataset aside (every in vivo dataset is in the supplemental material). Additionally, we specifically analyzed glucose clearance in the obesity model since we had suggestive data (please see from line 465 in the results). Therefore, we believe that we should examine this insight in a future study.

3. How about glucose uptake in ASK KO adipocyte in vitro?

Answer: 

Thank you for the question. We agree that there still be possible ASK1, 2, 3TKO adipocytes have a defect in glucose uptake in vitro; hence, we have measured glucose uptake levels of ASK TKO brown adipocytes with or without insulin stimulation. The results, shown in S1E Fig, suggest that ASK family does not engage in glucose uptake in mature brown adipocytes.

4. Gene names need to be italic. All characters in a protein name need to be capitalized.

Answer: 

Thank you for raising this issue. We have revised throughout the manuscript.

Additional changes

We have added Sho Sugawara as an author because of his contribution for the revised manuscript.

We have added spelled out abbreviations such as uncoupling protein 1 (UCP1) in line 47.

We have revised the reference because the original ref. 27 and 45 were the same.

We have revised a part of Western blotting method from line 146.

We have added methods of qRT-PCR and glucose uptake assay from line 170.

We have revised S3R and S Figs because the legends in the tables were wrongly denoted as S4.

We have corrected typos in the figures (S845 -> T845).

We have added a supplementary table for primer sequences.

---

## [Decision Letter · Decision Letter 1]

4 Sep 2020

β-adrenergic receptor signaling evokes the PKA-ASK axis in mature brown adipocytes

PONE-D-20-11137R1

Dear Dr. Hattori,

We’re pleased to inform you that your manuscript has been judged scientifically suitable for publication and will be formally accepted for publication once it meets all outstanding technical requirements.

Kind regards,

Makoto Kanzaki, Ph.D.

Academic Editor

PLOS ONE

Additional Editor Comments (optional):

Your revised manuscript was sent to the original referees, but one reviewer was unfortunately unable to review the revised manuscript. Upon careful reading by myself, I and another reviewer found that the revised manuscript has been much improved by addressing most of concerns raised by both reviewers. So I made this decision without sending it to new reviewer to avoid further delay. 

Reviewers' comments:

Reviewer's Responses to Questions

**Comments to the Author**

1. If the authors have adequately addressed your comments raised in a previous round of review and you feel that this manuscript is now acceptable for publication, you may indicate that here to bypass the “Comments to the Author” section, enter your conflict of interest statement in the “Confidential to Editor” section, and submit your "Accept" recommendation.

Reviewer #2: All comments have been addressed

2. Is the manuscript technically sound, and do the data support the conclusions?

Reviewer #2: Yes

3. Has the statistical analysis been performed appropriately and rigorously? 

Reviewer #2: Yes

4. Have the authors made all data underlying the findings in their manuscript fully available?

Reviewer #2: Yes

5. Is the manuscript presented in an intelligible fashion and written in standard English?

Reviewer #2: Yes

6. Review Comments to the Author

Reviewer #2: Responses are satisfactory. The manuscript has been improved properly and can be considered to be accepted.

7. PLOS authors have the option to publish the peer review history of their article (what does this mean?). If published, this will include your full peer review and any attached files.

Reviewer #2: No

---

## [Editor Report · Acceptance letter]

16 Oct 2020

PONE-D-20-11137R1 

β-adrenergic receptor signaling evokes the PKA-ASK axis in mature brown adipocytes 

Dear Dr. Hattori:

I'm pleased to inform you that your manuscript has been deemed suitable for publication in PLOS ONE. Congratulations! Your manuscript is now with our production department. 

Kind regards, 

on behalf of

Dr. Makoto Kanzaki 

Academic Editor

PLOS ONE